# A Unified Approach Towards Active Learning and Out-of-Distribution Detection

## Abstract

When applying deep learning models in real-world scenarios, active learning (AL) strategies are crucial for identifying label candidates from a nearly infinite amount of unlabeled data. In this context, robust out-of-distribution (OOD) detection mechanisms are essential for handling data outside the target distribution of the application. However, current works investigate both problems separately. In this work, we introduce SISOM as the first unified solution for both AL and OOD detection. By leveraging feature space distance metrics SISOM combines the strengths of the currently independent tasks to solve both effectively. We conducted extensive experiments showing the problems arising when migrating between both tasks. In these evaluations SISOM underlined its effectiveness by achieving first place in two of the widely used OpenOOD benchmarks and second place in the remaining one. In AL, SISOM [1] outperforms others and delivers top-1 performance in three benchmarks.

## 1 Introduction

Large-scale deep learning models encounter several data-centric challenges during training and operation, particularly in real-world problems such as mobile robotic perception. On the one hand, these models require vast amounts of data and labels for training, driven by the uncontrolled nature of real-world tasks. On the other hand, even when trained with extensive data, these models can behave unpredictably when encountering samples that deviate significantly from the training data, known as out-of-distribution (OOD) data.

Active learning (AL) addresses the first limitation by guiding the selection of label candidates. In the traditional pool-based AL scenario (Settles, 2010), models start with a small labeled training set and can iteratively query data and its labels from an unlabeled data pool. The selection is based on model metrics such as uncertainty, diversity, or latent space encoding. One AL cycle concludes with the model being trained on the labeled subset, including the newly added samples.

The second challenge, dealing with unknown data during operation, is typically addressed by OOD detection. OOD detection distinguishes between in-distribution (InD) data used for training the model and OOD samples, which differ from the training distribution. Literature differentiates between near-ODD and far-OOD, which can be categorized by the type of distribution shifts occurring. Yang et al. (2022b) assumes near-OOD as a pure covariate shift while far-OOD often contains a semantic shift.

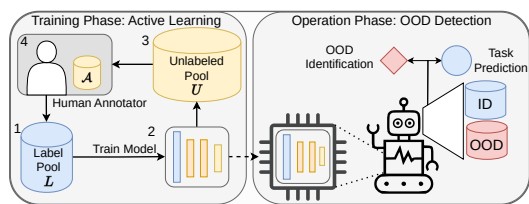

Figure 1: Real-world application life cycle comprising active learning in the training phase (left) and out-of-distribution detection in the operation phase (right).

Over the whole life cycle of mobile robotic applications, which consists of training and operation phases, both challenges occur. Fig. 1 illustrates such a life cycle with both tasks. Given an amount of collected data, AL is applied for a label-efficient training, while OOD detection is employed to control

---

[1]SISOM will be published upon acceptance - for review https://tinyurl.com/sisom-iclr

the operation state, which is necessary for real-world operation domains. Existing works address these challenges separately, which can lead to diverging goals of AL and OOD methods. Additionally, addressing these tasks separately introduces significant overhead, especially for deployment and development like hyperparameter optimization or the training of auxiliary models.

From a method perspective similarities between AL and OOD detection are even more evident, specifically both methodologies utilize common metrics, such as uncertainty, latent space distances, and energy. In addition, a sample detected by such metrics can be, on the one hand, a novel AL sample that is insufficiently represented by the current training distribution. On the other hand, the sample can pose a covariate shift in an OOD setting. Considering both cases as depicted in Fig. 2 show an ambiguity and overlap of both sample categories. This raises the question if an examination of the ambiguity and relation between the respective samples can provide valuable insights for designing approaches for both tasks.

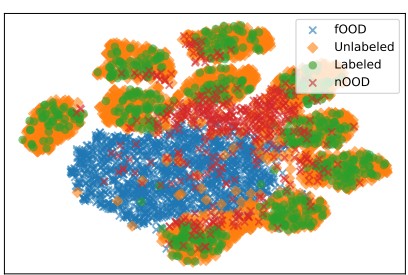

Figure 2: TSNE plot of unlabeled and OOD data compared to labeled data for CIFAR-10 as InD with 20% labeled, tiny ImageNet as near-OOD and SVHN as far-OOD.

*In our work*, we examine the connection between both tasks and design a novel approach by leveraging mutual strengths providing an effective solution for both tasks. Specifically, we employ enriched feature space distances based on neural coverage to propose **S**imultaneous **I**nformative **S**ampling and **O**utlier **M**ining (SISOM), which create a symbiosis between AL and OOD detection. By exploiting the ambiguity of both tasks, SISOM effectively archives *top-1* performance in most OOD benchmarks and, at the same time, surpasses existing AL methods with *top-1* performance. With its joint approach, SISOM provides an efficient simplification for application life cycles by *eliminating an additional OOD detection design phase* and avoiding conflicting design goals. Additionally, SISOM provides a *novel latent space analysis* for *post-training latent space refinement* and a first-of-its-kind *self-balancing of uncertainty and diversity metrics*.

In summary, *our contributions* are as follows:

- We explore the entanglement of AL and OOD detection.
- We propose **S**imultaneous **I**nformative **S**ampling and **O**utlier **M**ining, a novel method designed for both OOD detection *and* AL.
- We introduce a latent space analysis enabling an *optimization loop* for further *post-training latent space refinement* and a *self-balanced uncertainty diversity fusion*.
- In extensive experiments, we demonstrate SISOM effectiveness in AL *and* OOD benchmarks.

## 2 PRELIMINARIES

**Active Learning:** AL is a subfield of machine learning designed to reduce the number of required labels by querying a set of new samples $\mathbb{A}$ of a query size $q$ in a cyclic process. Let $\mathcal{X}$ represent a set of samples and $\mathcal{Y}$ a set of labels. AL starts with an initially labeled pool $\mathbb{L}$, containing data samples with features $\mathbf{x}$ and corresponding label $y$, and an unlabeled pool $\mathbb{U}$ where only $\mathbf{x}$ is known. However, $y$ can be queried from a human oracle. We further assume that $\mathbb{L}$ and $\mathbb{U}$ are samples from a distribution $\Omega$. In each cycle, a model $f$ is trained such that $f : \mathcal{X}_L \to \mathcal{Y}_L$. This model then selects new samples from $\mathbb{U}$ based on a query strategy $Q(\mathbf{x}, f)$, which utilizes (intermediate) model outputs. As a result, the newly annotated set $\mathbb{A}$ is added to the labeled pool $\mathbb{L}^{i+1}$ and removed from the unlabeled pool $\mathbb{U}^{i+1}$.

**Out of Distribution Detection:** Ancillary, OOD detection assumes a model $f : \mathcal{X}_L \to \mathcal{Y}_L$ trained on our training data $\{\mathbf{x}, y\} \in \mathbb{L}$ which have been sampled from the distribution $\Omega$. During evaluation or inference, a model $f$ encounters data samples $\tilde{\mathbf{x}}$ from a distribution $\Theta$ and $\Omega$, where $\Omega \cap \Theta = \emptyset$ and $\tilde{\mathbf{x}} \notin \mathbb{L}$. Data sampled from $\Omega$ are referred to as InD data, while samples from $\Theta$ are referred to as OOD data. Based on the trained model $f$, a metric $S$ is used to determine whether a sample $x$ is

sampled from $\Omega$ or $\Theta$.

$$G(\mathbf{x}, f) = \begin{cases} \text{InD} & \text{if } S(\mathbf{x}; f) \geq \lambda \\ \text{OOD} & \text{if } S(\mathbf{x}; f) < \lambda \end{cases} \tag{1}$$

OOD detection is further categorized into near- and far-OOD (Zhang et al., 2023). Far-OOD refers to completely unrelated data, such as comparing MNIST (LeCun et al., 1998) to CIFAR-100 (Krizhevsky et al., 2009), while CIFAR-10 (Krizhevsky et al., 2009) to CIFAR-100 would be considered as near-OOD. OpenODD (Yang et al., 2022b) ranks near-OOD detection as more challenging.

## 3 RELATED WORK

Given the disentanglement of fields, we review the related work individually.

**Active Learning:** AL mainly considers the pool-based and stream-based scenario (Settles, 2010), where data is either queried from a pool in a data center or a stream on the fly. For deep learning, the majority of current research deals with pool-based AL (Ren et al., 2021). However, further scenarios have been evaluated by Schmidt & Günnemann (2023) and Schmidt et al. (2024). Independent of the scenarios, samples are selected either by prediction uncertainty, latent space diversity, or auxiliary models. A majority of the uncertainty-based methods rely on sampling - like Monte Carlo Dropout (Gal & Ghahramani, 2016) - or employ ensembles (Beluch et al., 2018; Lakshminarayanan et al., 2017). To additionally ensure batch diversity Kirsch et al. (2019) used the joint mutual information. The uncertainty concepts have been employed and further developed for major computer vision tasks, including object detection (Feng et al., 2019; Schmidt et al., 2020), 3D object detection (Hekimoglu et al., 2022; Park et al., 2023), and semantic segmentation (Huang et al., 2018). One of the few works breaking the gap between both tasks (Shukla et al., 2022) modified an OOD detection method for pose estimation. Mukhoti et al. (2023) proposed an uncertainty baseline based on spectral convolutions and Gaussian mixture models, which shows effectiveness on AL and OOD detection compared to other uncertainty approaches. In contrast, diversity-based approaches aim to select key samples to cover the whole dataset. Sener & Savarese (2018) proposed to choose a CoreSet of the latent space using a greedy optimization. Yehuda et al. (2022) selected samples having high coverage in a fixed radius for low data regimes. Mishal & Weinshall (2024) extends the approach for more data regimes dynamic strategy mixing. Ash et al. (2020) enriched the latent space dimensions to the dimensions of the gradients and included uncertainty in this way. The concept of combining uncertainty with diversity has been further refined for 3D object detection (Yang et al., 2022a; Luo et al., 2023). Liang et al. (2022) combined different diversity metrics for the same task. In semantic segmentation, Surprise Adequacy (Kim et al., 2020) has been employed to measure how surprising a model finds a new instance. Besides the metric-based approach, the selection can also be made by auxiliary models mimicking diversity and uncertainty. These approaches range from loss estimation (Yoo & Kweon, 2019), autoencoder-based approaches (Sinha et al., 2019; Zhang et al., 2020; Kim et al., 2021) and graph models (Caramalau et al., 2021), to teacher-student approaches (Peng et al., 2021; Hekimoglu et al., 2024).

**Out-of-Distribution Detection:** To facilitate a fair comparison and evaluation of OOD methods, benchmarking frameworks like OpenOOD (Yang et al., 2022b; Zhang et al., 2023) have been introduced, which categorizes the methods into preprocessing methods altering the training process and postprocessing methods being applied after training. Preprocessing techniques include augmenting training data like mixing (Zhang et al., 2018; Tokozume et al., 2018) different samples or applying fractals to images (Hendrycks et al., 2022). Postprocessing approaches include techniques of manipulations on neurons and weights of the trained network, such as filtering for important neurons (Ahn et al., 2023; Djurisic et al., 2022), or weights (Sun & Li, 2022), or clipping neuron values to reduce OOD-induced noise (Sun et al., 2021). Logit-based approaches encompass the model output to estimate uncertainties using temperature-scaling (Liang et al., 2018), modified entropy scores (X. Liu, 2023), energy scores (Liu et al., 2020; Elflein et al., 2021) or ensembles (Arpit et al., 2022). Other methods rely on distances in the feature space, such as the Mahalanobis distance between InD and OOD samples (Lee et al., 2018), consider the gradients after a forwardpass (Liang et al., 2018; Hsu et al., 2020; Huang et al., 2021; Schwinn et al., 2021), estimate densities (Charpentier et al., 2020; 2022) or k nearest neighbor on latent space distances (Sun et al., 2022). A different branch operates on the features directly and evaluates properties like the Norm (Yu et al., 2023) or performs rank reductions via SVD (Song et al., 2022). NAC (Liu et al., 2024) combined gradient information with a density approach, where a probability density function over InD samples is estimated.

**OpenSet Active Learning:** The emerging field of OpenSet AL considers both tasks in one cycle, assuming the AL pool is polluted by OOD samples. Existing approaches (Ning et al., 2022; Park et al., 2022; Yang et al., 2023; Safaei et al., 2024) address both tasks with *separate* modules containing auxiliary models. None of the works investigates the correlation of AL and OOD samples. As both tasks are considered decouples with uncorrelated modules, this field is orthogonal to our examination of correlation and entanglement. We believe that this field profits from the joint consideration of AL and OOD samples as well as an examination of their ambiguity.

While various works exist in OOD and AL, both tasks are considered independent. Even in OpenSet AL, the tasks are considered by independent method components. Some uncertainty methods are evaluated on both tasks but limit their evaluation to the uncertainty domain. Current state-of-the-art approaches are often specified for one task. In addition, the application life cycle consideration is unexplored.

## 4 METHODOLOGY

To address both AL and OOD detection tasks in a unified method to simplify real-world applications, we need to first understand the goals of these two tasks. AL aims to identify and select samples that are beneficial for training and increase the models performance. These samples typically position themselves between the existing clusters in the latent space or near the decision boundaries. OOD detection targets the identification of data outside the training data and, therefore, outside the known clusters in latent space. Given the definition of far- and near-OOD, near-ODD is closer to InD data and located close to the decision boundaries and in between the existing clusters. Liu & Qin (2024) recently showed that OOD is generally closer to the decision boundary than InD confirming this hypothesis. Fig. 2 depicts this consideration showing the overlap of interesting unlabeled data and (near-)OOD data.

To target these overlapping regions we design a method focusing on the latent space regions between the clusters. To do so SISOM employs an enlarged feature space Coverage (1) and increases expressiveness by weighting important neurons in a Feature Enhancement (2). Based on this feature representation, we refine the AL selection and the InD and OOD border by using an inner-to-outer class Distance Ratio (3), guiding it to unexplored and decision boundary regions. As feature space distances are prone to poorly defined latent space representations, we introduce Feature Space Analysis (4) providing a self-deciding fusion of our distance metric with an uncertainty-based energy score. Optionally, our previous analysis enables us to optimize the Sigmoid Stepness (5), providing a further refinement of the feature space representations from (2). An overview is depicted in Fig. 3.

**(1) Coverage:** We aim to identify the regions of the samples that are interesting and unexplored for AL as well as OOD samples in latent space. To do so, we rely on an informative latent space covering as much information as possible.

To increase the information gain we cover the full network and define the feature space representation of an input sample $\mathbf{x}$ as a concatenation of the latent space of multiple layers $h_j$ in a set of selected layers $H$ in Eq. (2). This approach follows the procedures of neural coverage (Kim et al., 2019; Liu et al., 2024) and is contrasting to most diversity-based AL approaches (Sener & Savarese, 2018; Ash et al., 2020), which use a single layer.

$$\mathbf{z} = h_1(\mathbf{x}) \oplus \cdots \oplus h_j(\mathbf{x}) \oplus \cdots \oplus h_n(\mathbf{x}) \tag{2}$$

Given the feature space $\mathbf{z}$, we further denote $\mathbb{Z}_U$ as a set of feature space representations of unlabeled samples from $\mathbb{U}$, while $\mathbb{Z}_L$ denotes the set of representations of all labeled samples $\mathbb{L}$.

**(2) Feature Enhancement:** To enhance the expressiveness of our defined latent space we introduce a weighting of individual layers. Prior research (Huang et al., 2021; Liu et al., 2024) have demonstrated that the gradients of neurons with respect to the KL divergence of the model's output and a uniform distribution encapsulate valuable information for OOD detection.

We apply the technique to improve the features further and enrich these by representing the individual contribution of each neuron $i$, denoted as $g_i$. This gradient describes each neuron's contribution to the actual output being different from the uniform distribution. A low value suggests that the neuron has little influence on the prediction of a given input sample. Conversely, if the value is high, the respective neuron is crucial for the decision process.

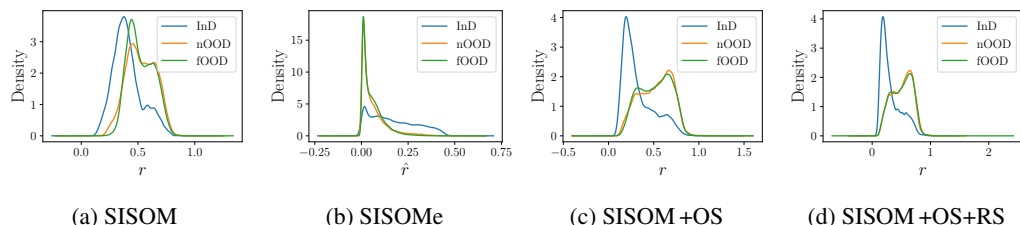

| (a) SISOM | (b) SISOMe | (c) SISOM +OS | (d) SISOM +OS+RS |

Figure 4: Density plots for SISOM with energy, Optimal Sigmoid Steepness (OS) and Reduced Subset Selection (RS) on CIFAR-100 with near-OOD (nOOD) and far-OOD (fOOD) as defined in OpenOOD.

Thus, the gradient vector can be interpreted as a saliency weighting for the activation values in the feature space to support seprability. In detail, we compute the gradient of the Kullback-Leibler (KL) divergence between an uniform distribution $u$ and the softmax output distribution $f(\mathbf{x})$ for an input $\mathbf{x}$:

$$\mathbf{g}_i = \frac{\partial D_{KL}(u||f(\mathbf{x}))}{\partial \mathbf{z}_i}. \qquad (3)$$

We incorporate the calculated saliency to create a weighted feature representation forming the enhanced feature space with the sigmoid function $\sigma$:

$$\tilde{\mathbf{z}} = \sigma(\mathbf{z} \odot \mathbf{g}). \qquad (4)$$

The resulting gradient-weighted feature representation effectively prioritizes the most influential neurons for each input. This facilitates the identification of inputs activating atypical influence patterns, which is significant for AL as well as OOD detection. A qualitative analysis demonstrating the effect of the feature enrichment is given in Appendix A.3.

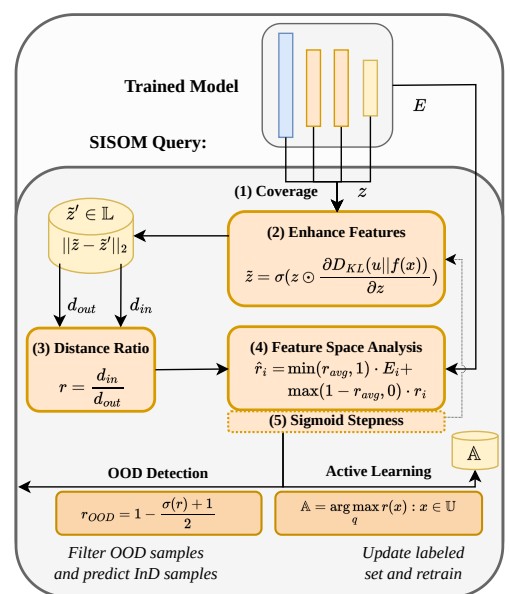

Figure 3: SISOM framework for OOD detection and AL combined.

**(3) Distance Ratio:** After we defined and enhanced our latent space we design our metric to identify the respective samples. Contrasting to other works in the latent space domain for AL and OOD detection (Sener & Savarese, 2018) which relay on simple distance metrics, we take inspiration from complex distance metrics (Kim et al., 2019) for detecting adversarial examples.

We assume the location of important samples in between the existing clusters in latent space. While samples closer to these clusters, like near-OOD or AL samples close to the decision boundary, are more important, far-OOD samples and exotic AL samples should not be omitted. To identify samples in these regions, we rely on a distance quotient between inner-class and outer-class distances.

The inner-class distance $d_{in}$ is defined as the minimal feature space distance to a known sample of the same class $c$ as the predicted pseudo-class of the given sample. The outer-class distance $d_{out}$ represents the minimal feature space distance to a known sample of a different class than the sample's pseudo-class.

$$d_{in} = \min_{\mathbf{z}' \in \mathbb{Z}_L(c'=c)} ||\tilde{\mathbf{z}} - \tilde{\mathbf{z}}'||_2 \qquad (5) \qquad d_{out} = \min_{\mathbf{z}' \in \mathbb{Z}_L(c' \neq c)} ||\tilde{\mathbf{z}} - \tilde{\mathbf{z}}'||_2 \qquad (6)$$

The distance is computed on the gradient-enhanced feature space $\tilde{\mathbf{z}}$ defined in Eq. (4) with $\mathbf{z}'$ describing the nearest sample from the set of known samples $Z_L$.

In many state-of-the-art works on AL, computationally expensive distance calculations are often present (Sener & Savarese, 2018; Ash et al., 2020; Caramalau et al., 2021). To make our approach

more efficient for AL and feasible for large-scale OOD detection tasks, we select a representative subset $\mathbb{T} \subset \mathbb{Z}_L$ as a comparison set, thereby significantly reducing computational overhead. We modify the Probcover (Yehuda et al., 2022) approach to select class-wise samples, maximizing coverage within a sphere with a fixed radius in the feature space. The effect of this subset selection is further investigated in Section 5.3.

Our SISOM score $r$ reflects the distance between each neuron's weighted feature representation in the latent space and the nearest sample of the predicted class relative to the closest distance to a sample from a different class:

$$r = \frac{d_{in}}{d_{out}}. \tag{7}$$

An extended comparison of the different distance metrics and their ability to separate InD and OOD is shown in Appendix A.3, while a SISOM is depicted in Fig. 4a.

For AL we select the $q$ samples with the highest distance ratio $r$, with $q$ being the AL query size:

$$\mathbb{A} = \text{argmax}_q r(\mathbf{x}) : \mathbf{x} \in \mathbb{U}. \tag{8}$$

For OOD Detection, we map the distance ratios $r$ to an interval $[0; 1]$ with the strictly monotonically decreasing function:

$$r_{OOD} = 1 - \frac{\sigma(r) + 1}{2}. \tag{9}$$

**(4) Feature Space Analysis:** Having a well-defined latent space is crucial for SISOM to attain optimal performance. Furthermore, we hypothesize that techniques relying on feature space metrics are more dependent on feature space separation than uncertainty-based methods. This dependency is important for SISOM as it utilizes a quotient of feature space metrics. Nevertheless, obtaining a well-defined and separable latent space may pose challenges in specific contexts and tasks.

To estimate the separability of feature space, we compute the average distance ratio $r_{avg}$ using Eq. (4) and Eq. (7) for the known set as:

$$r_{avg} = \frac{1}{|L|} \sum_{\tilde{\mathbf{z}} \in \mathbb{L}} \frac{d_{in}(\tilde{\mathbf{z}})}{d_{out}(\tilde{\mathbf{z}})} = \frac{1}{|L|} \sum_{\mathbf{z} \in \mathbb{L}} \frac{d_{in}(\sigma(\mathbf{z} \odot \mathbf{g}))}{d_{out}(\sigma(\mathbf{z} \odot \mathbf{g}))}. \tag{10}$$

A lower $r_{avg}$ value indicates better separation of the samples in the enhanced feature space, implying that samples of the same class are relatively closer together than samples of different classes. To mitigate possible performance disparities of SISOM in difficult separable domains, we introduce a novel self-deciding process for the sampling method, which utilizes the feature separation score $r_{avg}$ as follows:

$$\hat{r}_i = \min(r_{avg}, 1) \cdot E_i + \max(1 - r_{avg}, 0) \cdot r_i. \tag{11}$$

The so created $\hat{r}$ combines our SISOM score from Eq. (7) with the uncertainty-based energy score $E(\mathbf{x}) = -\log \sum_{i=1}^{c} \exp(f(\mathbf{x})_i)$ based on the model's output logits $f(\mathbf{x})$.

Depending on whether $r_{avg} \to 1$ or $r_{avg} \to 0$, the created score $\hat{r}_i$ relies more on either the energy score or the distance ratio $r_i$. If $r_{avg} \to 1$, indicating poorly separated classes, $\hat{r}_i$ relies more on the energy score. Conversely, if $r_{avg} \to 0$, suggesting a well-separated feature space, $\hat{r}_i$ relies more on the distance ratio. A density outline of our combined approach SISOMe is given in Fig. 4b. Alternatively, one can replace $r_{avg}$ with a tuneable hyperparameter in Eq. (11).

**(5) Sigmoid Steepness:** Since Eq. (10) depends on the sigmoid function defined in Eq. (4), the sigmoid function has a large influence on the enhanced feature space $\tilde{\mathbf{z}}$. An additional hyperparameter $\alpha$ can influence the sigmoid function's steepness. As $\mathbf{z}$ is concatenated from different layers in Eq. (2), the sigmoid can be applied to each layer $j$ individually. This allows for a more nuanced control over the influence of each neuron's contribution to the final decision and so influences the separability of the feature space. We define the sigmoid using the steepness parameter $\alpha$ as:

$$\sigma_j(\mathbf{x}) = \frac{1}{1 + e^{-\alpha_j \mathbf{x}}}; \quad \{\alpha_j : h_j \in \mathbf{z} \ \forall j\}. \tag{12}$$

Relating to Eq. (4), the set $\alpha$ of steepness parameters of the sigmoid function for each layer $h_j$, determines the degree of continuity or discreteness of the features within that layer. By applying a

layerwise sigmoid, Eq. (4) is formulated as follows:

$$\tilde{\mathbf{z}} = \sigma_1(h_1(\mathbf{x}) \odot g_{i,1}) \oplus \cdots \oplus \sigma_j(h_j(\mathbf{x}) \odot \mathbf{g}_{i,j}) \oplus \tag{13}$$
$$\cdots \oplus \sigma_n(h_n(\mathbf{x}) \odot \mathbf{g}_{i,n}),$$
$$\text{with} \quad \mathbf{g}_{i,j} = \frac{\partial D_{KL}(u||f(\mathbf{x}))}{\partial h_{j,i}}; \quad \forall j.$$

Following this consideration we can select $\alpha$ values which optimize the feature space separability metric $r_{avg}$ from Eq. (10) by minimizing $\alpha_{opt} = \arg\min_\alpha r_{avg}(\alpha)$. Besides the quantitative assessment of our Feature Space Analysis and Sigmoid Steepness in Section 5.3, the influence of the Sigmoid Steepness is shown in Fig. 4c.

## 5 EXPERIMENTS

To evaluate our proposed method, we conducted a comprehensive assessment of SISOM on both tasks AL and OOD detection individually. We consider compound tasks like Openset AL as out of scope as existing works address the sub-task by individual components, while SISOM showcases the ambiguity of both task sample characteristics. The experiments' details, settings, and results are presented in Section 5.1 and Section 5.2, respectively. We further conduct an ablation study in Section 5.3. We utilized the standard pool-based AL scenario (Settles, 2010) for AL. For OOD detection, we followed the widely used OpenOOD benchmarking framework (Yang et al., 2022b; Zhang et al., 2023).

In the AL experiments, we compared our method against several baselines, including **CoreSet** (Sener & Savarese, 2018), **CoreGCN** (Caramalau et al., 2021), **Random**, **Badge** (Ash et al., 2020), and **Loss Learning** (Yoo & Kweon, 2019). Additionally, we adapted the **NAC** (Liu et al., 2024) method from OOD detection to AL to assess the transferability from OOD to AL.

For OOD detection experiments, we employed the implementation provided by the OpenOOD framework when available. We also followed the experimental setup and datasets for near- and far-OOD detection. The baselines used for validation include **NAC** (Liu et al., 2024), **Ash** (Djurisic et al., 2022), **KNN** (Sun et al., 2022), **Odin** (Hsu et al., 2020), **ReAct** (Sun et al., 2021), **MSP** (Hendrycks & Gimpel, 2016), **Energy** (Liu et al., 2020), **Dice** (Sun & Li, 2022), **RankFeat** (Yu et al., 2023), **FeatureNorm** (Song et al., 2022) and **GEN** (X. Liu, 2023). Moreover, we tested the **CoreSet** (Sener & Savarese, 2018) AL method to verify the transferability from AL to OOD. Our focus was on methods that use the cross-entropy training scheme to maintain a fair comparison and ensure compatibility post-AL.

### 5.1 ACTIVE LEARNING

We followed the most common AL benchmark settings and datasets, including the CIFAR-10 (Krizhevsky et al., 2009), CIFAR-100 (Krizhevsky et al., 2009), and SVHN (Netzer et al., 2011) datasets paired with a ResNet18 (He et al., 2016) model. We assessed the network's performance by measuring accuracy relative to the amount of data used. The plots include markers to indicate the selection steps. As suggested by (Yoo & Kweon, 2019; Ash et al., 2020), we start with an initial pool size of 1,000 labeled samples for CIFAR-10 and SVHN. In each AL cycle, the model can query 1,000 additional samples from an unlabeled pool, which are then labeled and added to the labeled pool for the subsequent cycle. Due to the larger number of classes in CIFAR-100, we increased the selection size to 5,000. Detailed parameters and settings are available in Appendix B.1.

In the CIFAR-10 benchmark depicted in Fig. 5a, SISOM exhibits swift progress and maintains consistent performance from the outset. It consistently outperforms other methods, achieving the highest performance differential in all selection cycles and is only eclipsed by SISOMe especially in early cycles. Furthermore, as the sample size increases, our method maintains its superiority over Learning Loss and CoreSet. NAC does not demonstrate superior performance compared to Random.

After examining SISOM in datasets with a limited number of classes, we examine the AL setup on the larger CIFAR-100 dataset and report the results in Fig. 5b. In this setting, all methods are less stable in its ranking compared to the other dataset, reflecting the increased difficulty of the dataset. The complexity of the dataset requires more data for the model to perform effectively. While in the

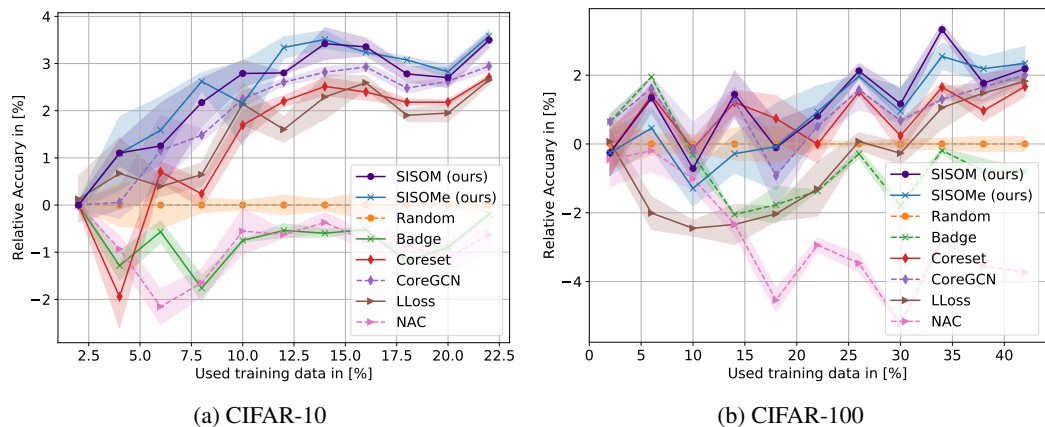

(a) CIFAR-10           (b) CIFAR-100

Figure 5: Comparison of different active learning methods on CIFAR-10, SVHN and CIFAR-100 with indicated standard errors.

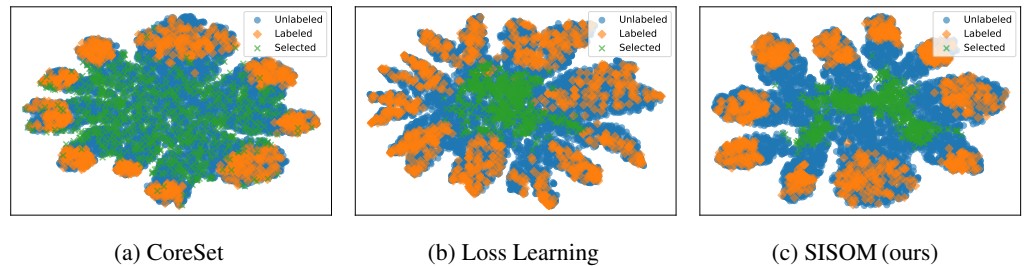

(a) CoreSet          (b) Loss Learning          (c) SISOM (ours)

Figure 6: T-SNE feature space comparison of Loss Learning, CoreSet and SISOM for SVHN on cycle 1. SISOM effectively targets the areas in-between the clusters.

early stages, pure diversity-based methods are in the lead, SISOM gains velocity in the last selection steps and achieves the highest performance difference only in the last step SISOMe is more effective.

Following the experiments on CIFAR-10 and CIFAR-100 we conducts experiments on SVHN and report them in Appendix A.1.

In conclusion of the AL experiments, SISOM reached state-of-the-art performance and surpasses other methods across all three datasets, demonstrating its viability for AL. While in the early stages, SISOM falls behind other approaches for CIFAR-100, in following selection cycles with more training data it outperforms them. We hypothesize that the early cycles had a poorly separated feature space, causing this issue.

## 5.2 OUT-OF-DISTRIBUTION DETECTION

Following our evaluation of SISOM on classic AL benchmarks, we utilize the OpenOOD framework to evaluate its performance on the OOD detection task. We stick to the recommended benchmarks on CIFAR-10 (Krizhevsky et al., 2009), CIFAR-100 (Krizhevsky et al., 2009), and ImageNet 1k (Deng et al., 2009), and we provide evaluation values for both near- and far-OOD detection. The assignment of datasets to near and far categories follows the framework's suggestions and is reported with additional settings in Appendix B.2. In addition, we benchmarked the life cycle setting in Appendix A.2. The framework ranks methods based on their AUROC performance and provides checkpoints for fair post-processor validation.

Firstly, we examine the performance on the CIFAR-10 benchmark and show the results in Table 1a. SISOMe and SISOM achieve the highest AUROC score for near-OOD data, respectively. SISOMe surpasses SISOM in all metrics. For far-OOD, SISOM ranks third after NAC, while SI-SOMe secures the first place. This is noteworthy as NAC underperformed in the AL task, even when compared to methods suffering from batch diversification, which underlines the non-triviality of migrating between both tasks out of the box.

Table 1: OOD benchmark for CIFAR-10, CIFAR-100 and ImageNet1k with Cross-Entropy training setting and dataset according to OpenOOD sorted by Near-OOD performance.

|  | (a) CIFAR-10 | | | | (b) CIFAR-100 | | | | (c) ImageNet 1k | | |
| Post-processor | OOD AUROC | | ID Acc. | Post-processor | OOD AUROC | | ID Acc. | Post-processor | OOD AUROC | | ID Acc. |
| | Near-OOD | Far-OOD | | | Near-OOD | Far-OOD | | | Near-OOD | Far-OOD | |
|---|---|---|---|---|---|---|---|---|---|---|---|
| SISOMe | **91.76** | **94.74** | 95.06 | Gen | **81.31** | 79.68 | 77.25 | SISOMe | **78.59** | 89.04 | 76.18 |
| SISOM | 91.40 | 94.50 | 95.06 | SISOMe | 80.96 | 79.8 | 77.25 | ASH | 78.17 | **95.74** | 76.18 |
| NAC | 90.93 | 94.60 | 95.06 | Energy | 80.91 | 79.77 | 77.25 | ReAct | 77.38 | 93.67 | 76.18 |
| KNN | 90.64 | 92.96 | 95.06 | ReAct | 80.77 | 80.39 | 77.25 | SISOM | 77.33 | 88.01 | 76.18 |
| CoreSet | 90.34 | 92.85 | 95.06 | MSP | 80.27 | 77.76 | 77.25 | GEN | 76.85 | 89.76 | 76.18 |
| GEN | 88.20 | 91.35 | 95.06 | KNN | 80.18 | 82.4 | 77.25 | KLM | 76.64 | 87.6 | 76.18 |
| MSP | 88.03 | 90.73 | 95.06 | ODIN | 79.9 | 79.28 | 77.25 | Energy | 76.03 | 89.50 | 76.18 |
| Energy | 87.58 | 91.21 | 95.06 | SISOM | 79.42 | 77.91 | 77.25 | MSP | 76.02 | 85.23 | 76.18 |
| ReAct | 87.11 | 90.42 | 95.06 | DICE | 79.38 | 80.01 | 77.25 | ODIN | 74.75 | 89.47 | 76.18 |
| FeatureNorm | 85.52 | 95.59 | 95.06 | ASH | 78.2 | 80.58 | 77.25 | DICE | 73.07 | 90.95 | 76.18 |
| ODIN | 82.87 | 87.96 | 95.06 | KLM | 76.56 | 76.24 | 77.25 | NAC | 71.73 | 94.66 | 76.18 |
| RankFeat | 79.46 | 75.87 | 95.06 | CoreSet | 75.69 | 79.53 | 77.25 | KNN | 71.1 | 90.18 | 76.18 |
| KLM | 79.19 | 82.68 | 95.06 | NAC | 72.00 | **86.56** | 77.25 | FeatureNorm | 67.57 | 91.13 | 76.18 |
| DICE | 78.34 | 84.23 | 95.06 | RankFeat | 61.88 | 67.10 | 77.25 | RankFeat | 50.99 | 53.93 | 76.18 |
| ASH | 75.27 | 78.49 | 95.06 | FeatureNorm | 47.87 | 80.99 | 77.25 | Coreset | - | - | 76.18 |

In the OpenOOD CIFAR-100 benchmark Table 1b the best far-OOD method shows the worst near-OOD performance, while for CIFAR-10, methods performed almost equally well on both near- and far-OOD. SISOMe ranks as the second-best method for near-OOD and repeatedly beats the individual metrics, SISOM and Energy. This is an interesting finding since, in contrast to CIFAR-10, energy achieves better performance than SISOM among the individual metrics on CIFAR-100. This supports our hypothesis that by considering the average ratio $r_{avg}$ as a proxy for feature space separation, we obtain stronger performances in both well-separated and poorly-separated feature spaces.

The third benchmark suggested by OpenOOD is ImageNet 1k, which contains more classes and is a much larger dataset than the previous ones. In the results depicted in Table 1c, SISOMe and SISOM achieved first and fourth-best scores on near-OOD, with SISOMe showing strong performance for far-OOD. Interestingly, the NAC method, which was the second-best in CIFAR-10, ranks much lower, and KNN, the third-best method in CIFAR-10, ranks last. Meanwhile, ASH, which ranks first in this benchmark, is last in the CIFAR-10 benchmark.

To evaluate the life cycle perspective we conducted additional experiments using the AL models in A.2.

Overall benchmarks, SISOMe is the only approach, being consistently under the top three ranks, and even secured first place in two of them. Excluding SISOMe , SISOM achieved one top-three ranking and one top-one ranking. Notably, our method performs relatively better on near-OOD data than on far-OOD data. This is understandable, as the ratio between inner and outer class distance is higher for data close to the training data distribution, while the quotient is lower for far-OOD. Additionally, near-OOD is closer to the data of interest for AL selection. According to (Yang et al., 2022b), near-OOD is considered the more challenging task and is more likely to occur in real-world applications. Thus, higher performance on near-OOD may be preferred in practice.

## 5.3 ABLATIONS STUDIES

In an ablation study, we qualitatively examine the latent space assumptions for AL as well as the effect of unsupervised feature space analysis and reduce labeled set $\mathbb{T}$. A study of the individual components of SISOM is given in Appendix A.3.

**AL Latent Space:** To validate the assumptions made in Section 4, we examine the configuration of the latent space of our selection in the AL experiments. The objective of our method is to select samples in the decision boundary region for the AL case. In Fig. 6, we compare CoreSet and Loss Learning with SISOM . It can be observed that CoreSet, as intended, exhibits high diversity in unseparated regions. The pseudo-uncertainty-based Loss Learning method is more concentrated in its selection but fails to diversify the selection across all decision boundaries. In contrast, SISOM , as shown in Fig. 6c, focuses on the decision boundary while successfully covering the entire area between the unseparated samples. This demonstrates the effectiveness of our method in addressing the challenges of both AL and OOD detection.

**Optimal Sigmoid Steepness:** In our feature space analysis in Section 4, we derived $r_{\text{avg}}$ in Eq. (10) as a proxy for the feature space separability. Due to the distance concept of SISOM , we hypothesize that it works better in well-separated feature spaces. To examine this, we conduct a random search for different $\alpha$ sets and record the different $r_{\text{avg}}$ values. To reduce the search space, we follow the premise postulated in Section 4 that generally, deeper layers require a steeper sigmoid curve, i.e., a higher $\alpha_j$ value due to the nature of the features captured within these layers.

After computing every $r_{\text{avg}}$ value for each combination of $\alpha$, we select the $\alpha_{\text{opt}}$ set that minimizes $r_{\text{avg}}$. Formally, this can be written as:

$$\alpha_{\text{opt}} = \arg \min_{\alpha} r_{\text{avg}}(\alpha)$$

In Table 2, an optimized set $\alpha_{opt}$ is marked with OS. As it can be seen, a set with better feature space separation leads to increased performance for CIFAR-100 and ImageNet, partly confirming our hypothesis. In CIFAR-10 however, the original set of parameters yields the best results. One explanation might be that, in CIFAR-10, the different classes are already well separated, such that optimization on this separation yields no improvement and leads to an overfitting behavior.

**Reduced Subset Selection:** For larger datasets, distance-based approaches like CoreSet (Sener & Savarese, 2018) or (Ash et al., 2020) suffer from huge computational efforts, which is problematic for OOD detection, too. In Section 4, we suggested to use a reduced subset $\mathbb{T}$ of the comparison set $\mathbb{Z}_L$, selecting class-wise samples with the most neighbors in a given radius. For each dataset, we select a total of 10% of the samples for each class, drastically increasing inference speed. We compare the effect of our reduced subset selection (RS) in Table 2 and highlight it qualitatively in Fig. 4d. A comparison of the preprocessing steps for SISOM in Table 2 indicates that the AUROC near-OOD score has improved for all datasets. It can be observed that preselection enhances feature space separability based on the $r_{\text{avg}}$ column. This also strengthens our hypothesis from the previous subsection. For ImageNet and CIFAR-100, the combination of feature analysis and preselection results in the best performance, for CIFAR-10 the additional feature space analysis did not improve the performance. By taking the low $r_{\text{avg}}$ into account, the chosen values could have reduced the space too much, leading to an overfitting behavior. All parameters are given in Appendix B.3.

Table 2: Ablation Study on Optimal Sigmoid Steepness (OS) and Reduced Subset Selection (RS) on Near OOD Benchmarks.

| Method | ImageNet | | CIFAR 100 | | CIFAR 10 | |
|---|---|---|---|---|---|---|
| | $\text{AUROC}_n$ | $r_{\text{avg}}$ | $\text{AUROC}_n$ | $r_{\text{avg}}$ | $\text{AUROC}_n$ | $r_{\text{avg}}$ |
| SISOM | 77.21 | 0.270 | 75.93 | 0.33 | 91.33 | 0.26 |
| SISOM , OS | 77.4 | 0.266 | 79.56 | 0.19 | 90.37 | 0.099 |
| SISOM , RS | 77.33 | 0.249 | 76.07 | 0.31 | **91.40** | 0.24 |
| SISOM , OS, RS | **77.37** | 0.245 | **79.69** | 0.18 | 90.54 | 0.086 |

# 6 CONCLUSION

We proposed SISOM , the first approach designed to solve OOD detection and AL jointly, providing an effective simplification in real-world application life cycles by eliminating an OOD design phase and avoiding conflicting goals of AL and OOD detection. By weighting latent space features with KL divergence of the neuron activations and relating them to the latent space clusters of the different classes SISOM achieves state-of-the-art performance in both tasks. In addition, SISOM provides a novel feature space analysis scheme enabling a post-training feature space refinement as well as a self-guided uncertainty and diversity fusion introduced as SISOMe . In the famous OpenOOD benchmarks SISOM archives the *top-1* performance in *two of the three benchmarks* and the second place in the remaining one. For active learning, SISOM surpasses state-of-the-art approaches in three different benchmarks. While current state-of-the-art approaches are highly specialized for either AL or OOD detection, SISOM solves both tasks with the same approach. Underlined by these results, SISOM effectively addresses real-world applications, like environment sensing, which usually suffers from label costs during training and high unlabeled data availability as well as out-of-distribution samples during inference.

In future work, we plan to combine the two tasks that are currently separated as independent steps. Enabling continuous AL during inference while filtering out-of-distribution data can significantly enhance the model's performance after the initial selection phase.

## REPRODUCEABILTY STATEMENT

To ensure reproducibility, we conducted all experiments with the same fixed seeds, which are reported in the training procedure in the appendix. We used the exact parameter setting of the OpenOOD benchmark for the OOD experiments. Moreover, the code is released in the benchmark form with available configurations.

## ETHICS STATEMENT

With our research, we address the challenges of real-world and mobile (robotic) applications. While the common usage of robots or real-world applications does not pose ethical concerns, these fields pose the risk of misuse.

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

## A    ADDITIONAL EXPERIMENTS

In this section, we present additional experiments for SISOM .

### A.1    ACTIVE LEARNING - SVHN

Following the CIFAR experiments settings, we depict the results for the SVHN experiments in Fig. 7. Similar to the CIFAR-10 results, our method maintained high performance, but the method differences shrink with the easier the dataset. In the last cycle, SISOM reaches the highest performance, with a margin over other methods. As for CIFAR-10, NAC did not perform well in the data selection. Given that SVHN's 10 classes are numbers, it is easier than the more diverse CIFAR-10 benchmark dataset. This can be observed by an overall reduced performance gap between the methods compared to CIFAR-10.

### A.2    OUT-OF-DISTRIBUTION LIFE CYCLE

To evaluate the effectiveness of SISOM in a life cycle setting, we utilized the models after the AL cycle for an OOD benchmark. In Table 3 we used the same setting as for the benchmark CIFAR-10 experiments with the similar near- and far-OOD. It should be noted that while openOOD is open to deploy different checkpoints, modifying the InD data access is more challenging and remains unchanged. In Table 3 SISOMe archived the top performance, making it suitable for the full application life cycle.

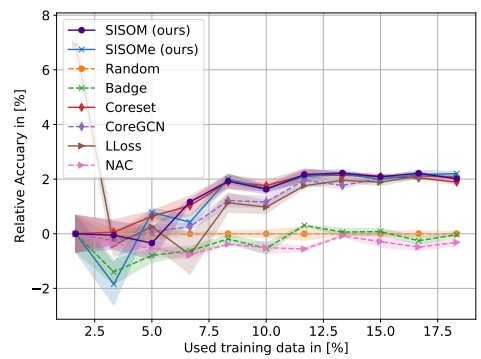

Figure 7: Comparison of different active learning methods on SVHN with indicated standard errors.

Table 3: OOD benchmark for CIFAR-10 using the AL checkpoints of SISOM .

| Postprocessor | OOD AUROC | | ID |
| | Near-OOD | Far-OOD | Acc. |
| --- | --- | --- | --- |
| SISOMe | **86.84** | **88.39** | 89.73 |
| ReAct | **86.84** | 87.72 | 89.73 |
| GEN | 85.43 | 86.04 | 89.73 |
| MSP | 84.37 | 84.85 | 89.73 |
| ASH | 83.39 | 87.33 | 89.61 |
| NAC | 82.26 | 85.06 | 89.73 |
| RankFeat | 60.20 | 56.73 | 60.84 |

### A.3    FEATURE SPACE ASSIGNMENTS

In this section, we highlight the influence of major components of our methods on the ability to separate InD and OOD data. In Fig. 8, we display the influence of the KL divergence gradient with a T-SNE analysis on CIFAR-10 (Krizhevsky et al., 2009) as InD and Tiny ImageNet (tin) (Le & Yang, 2015) as near-OOD. Without feature enhancement, the latent space is much harder to separate, and tin is distributed all over the latent space as shown in Fig. 8a. In contrast, the latent space with KL divergence enhances features, is much more separated, and has a clearer decision boundary to the near classes as indicated in Fig. 8b.

In addition to the previously presented density plots, we show the inner and outer distance together with the distance quotient of SISOM in Fig. 9 for CIFAR-10. Fig. 9a shows the inner class, indicating small inner class distances leading to a good separability for the InD data. On the other hand, the outer class distance in Fig. 9b provides a good separable peak for InD data, but a portion of InD overlaps with OOD data. The combined distance quotient shows the increased separability of the different InD and OOD sets as depicted in Fig. 9c.

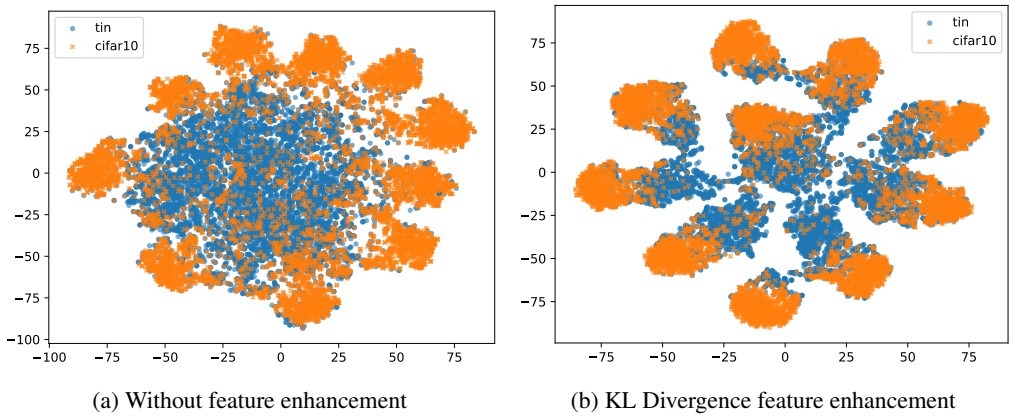

(a) Without feature enhancement       (b) KL Divergence feature enhancement

Figure 8: T-SNE comparison of the latent space for OOD detection with and without KL-Divergence feature enrichment.

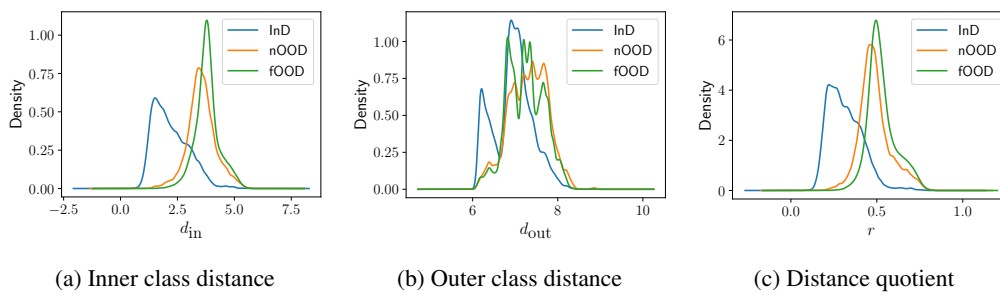

(a) Inner class distance      (b) Outer class distance      (c) Distance quotient

Figure 9: Density plots for the inner class distance, outer class distance, and the distance quotient of SISOM for CIFAR-10 with near-OOD (nOOD) and far-OOD (fOOD) as defined in OpenOOD..

## B  EXPERIMENTAL DETAILS

In this section, we provide experiment details to support the reproducibility of results by providing the used parameters[2].

### B.1  ACTIVE LEARNING EXPERIMENTS

In active learning experiments, we used a ResNet18 (He et al., 2016) model, with the suggested modifications of (Yoo & Kweon, 2019) presented in a CIFAR benchmark repository (kuangliu, 2021), which replaced the kernel of the first convolution with a $3 \times 3$ kernel. Additionally, we used an SGD optimizer with a learning rate of 0.1 and multistep scheduling at 60, 120, and 160, decreasing the learning rate by a factor of 10, which are reported benchmark parameters for CIFAR-100 (weiaicunzai, 2022). For SVHN and CIFAR-10 we used a learning rate of 0.025 and a cosine scheduler as suggested by Yehuda et al. (2022). For the construction of the feature space, we used the layers after the 4 blocks of ResNet with the following sigmoid values:

- **CIFAR-10**
  Adaptive Average Pooling Layer: 50,
  Sequential Layer 3: 10,
  Sequential Layer 2: 1,
  Sequential Layer 1: 0.05.

- **CIFAR-100/SVHN**
  Adaptive Average Pooling Layer: 1,

---

[2]Code will published upon acceptance, for review `https://tinyurl.com/sisom-iclr`

Sequential Layer 3: 0.1,
Sequential Layer 2: 0.1,
Sequential Layer 1: 0.1

## B.2 OUT-OF-DISTRIBUTION EXPERIMENTS

In the OOD experiments, we report the mean of the three different seeds employed in the standard setting of the OpenOOD (Yang et al., 2022b) framework with ResNet18 for CIFAR-10 and CIFAR-100. For Imagenet, we use the sole ResNet50 torchvision checkpoint provided in the standard settings. We utilized the near- and far-OOD assignments suggested by the benchmark listed below. We followed the official tables of OpenOOD's benchmark and reported the mean without the standard deviation. For the CIFAR-100 experiment, instead of using the automated $r_{avg}$ value to balance between $r$ and $E$ from Eq. (11), we set $r_{avg} = 0.8$ for SISOMe based on a hyperparameter study. In the benchmark tables, we reported for SISOM the best values matching the best values of the ablation study modifications. Furthermore, we follow the suggested sigmoid values (Liu et al., 2024) for CIFAR-10 and ImageNet. For CIFAR-100, we choose values that minimize Eq. (10). A detailed overview of the sigmoid values for the 4 blocks of ResNet18 and ResNet50 for all experiments is provided below:

- **CIFAR-10**
  Adaptive Average Pooling Layer: 100,
  Sequential Layer 3: 1000,
  Sequential Layer 2: 0.001,
  Sequential Layer 1: 0.001

- **CIFAR-100**
  Adaptive Average Pooling Layer: 1,
  Sequential Layer 3: 0.1,
  Sequential Layer 2: 0.1,
  Sequential Layer 1: 0.1

- **ImageNet**:
  Adaptive Average Pooling Layer: 3000,
  Sequential Layer 3: 300,
  Sequential Layer 2: 0.01,
  Sequential Layer 1: 1

OOD dataset assignment:

- **CIFAR-10**
  Near-OOD: CIFAR-100 (Krizhevsky et al., 2009), Tiny ImageNet (Le & Yang, 2015)
  Far-OOD: MNIST (LeCun et al., 1998), SVHN (Netzer et al., 2011), Textures (Cimpoi et al., 2014), Places365 (López-Cifuentes et al., 2020)

- **CIFAR-100**
  Near-OOD: CIFAR-10 (Krizhevsky et al., 2009), Tiny ImageNet (Le & Yang, 2015)
  Far-OOD: MNIST (LeCun et al., 1998), SVHN (Netzer et al., 2011), Textures (Cimpoi et al., 2014), Places365 (López-Cifuentes et al., 2020)

- **ImageNet**
  Near-OOD: SSB-hard (Vaze et al., 2021), NINCO (Bitterwolf et al., 2023)
  Far-OOD: iNaturalist (Van Horn et al., 2018), Textures (Cimpoi et al., 2014), OpenImage-O (Wang et al., 2022)

## B.3 ABLATION STUDY

In this section, we highlight the relevant parameters for the ablation study experiments on SISOM. Namely, we examine the Optimal Sigmoid Steepness (OS) and the Reduced Subset Selection (RS) shown in Tab. 4. In the experiments conducted with RS, a representative subset size of 10% relative to the original training set was used across all experiments. Additionally, the specific distance radius used for the class-wise ProbCover (Yehuda et al., 2022) implementation on CIFAR-10, CIFAR-100, and ImageNet is provided in Table 4. For SISOM + RS without OS, the suggested sigmoid values

Table 4: Parameters for the Ablation Study, Probcover Radius for RS and Search Space of Optimal Sigmoid Steepness.

| Dataset | ProbCover Radius | Layer | Sigmoid Search Values |
|---------|------------------|-------|-----------------------|
| CIFAR-10 | 0.75 | AdaptiveAvgPool2d-1 | **100**, 1000 |
| | | Sequential-3 | **1**, 10, 1000 |
| | | Sequential-2 | **0.001**, 0.1, 1 |
| | | Sequential-1 | **0.001**, 0.1, 1 |
| CIFAR-100 | 5.0 | AdaptiveAvgPool2d-1 | **1**, 50, 100 |
| | | Sequential-3 | **0.1**, 10, 100 |
| | | Sequential-2 | **0.1**, 1 |
| | | Sequential-1 | 0.005, **0.1** |
| ImageNet | 10.0 | AdaptiveAvgPool2d-1 | 10, 100, **3000** |
| | | Sequential-3 | **1**, 10, 300 |
| | | Sequential-2 | **0.1**, 1 |
| | | Sequential-1 | **0.1**, 1 |

(Liu et al., 2024) emphasized in Appendix B.2 were used. For the OS modification, the search space for the optimal sigmoid parameters is presented in Table 4. The parameters fulfilling the minimization of Eq. (10) are highlighted in bold.

