# OpenReview forum: "A Unified Approach Towards Active Learning and Out-of-Distribution Detection"
_ICLR.cc/2025/Conference — ICLR 2025 Conference Withdrawn Submission_

### Official Review · Reviewer_UD7a · 2024-10-27

**Soundness:** 1
**Presentation:** 2
**Contribution:** 1
**Rating:** 1
**Confidence:** 5

**Summary:**

This paper introduces SISOM (Simultaneous Informative Sampling and Outlier Mining), a novel approach that addresses both active learning (AL) and out-of-distribution (OOD) detection in a unified framework. By leveraging feature space distance metrics, SISOM combines the strengths of these previously independent tasks and demonstrates effectiveness in extensive experiments. The method achieved top performance in multiple OOD benchmarks and surpassed existing AL methods in benchmark tests. Additionally, SISOM introduces a new latent space analysis for post-training refinement and a self-balancing fusion of uncertainty and diversity metrics.

**Strengths:**

1. The paper provides a clear and detailed description of the SISOM method, making it accessible and understandable.
2. The authors conducted extensive experiments across various datasets and compared the method against multiple benchmarks, substantiating SISOM's efficacy.

**Weaknesses:**

1. The problem addressed in this paper is essentially consistent with existing open-set active learning (OSAL) methods [1-7]. However, the authors provide a highly inappropriate justification for separating the two, allowing their proposed method to avoid comparison with these existing methods. In OSAL tasks, the goal of AL is to query known class samples with high information content. The authors' assertion that "existing methods address AL and OOD separately" is misleading, as it merely represents one specific means to achieve this goal. In OSAL, unknown class samples will inevitably be queried, leading to the presence of unknown class samples in the labeled set. Not utilizing these samples is a waste of labeling resources. Since directly inputting unknown class samples into the target classifier to train a k+1 classifier often harms the performance of known classes, introducing an additional classifier to assist in active querying is a natural idea. So, the authors' claim that existing methods incorporate auxiliary models for differentiation is also unreasonable. Furthermore, not all existing methods utilize auxiliary models, such as CCAL [1], and those that do can often have the auxiliary models removed with appropriate modifications. After all, introducing auxiliary models alone is insufficient for these papers to be accepted; each has its own unique innovations to achieve the goals of OSAL.

2. The contributions in the methods section are negligible. The authors' approach of distinguishing samples based on their distances in the feature space is a common practice, with similar methods already proposed in [5, 6].

3. Regarding the experimental section, the active learning method chosen by the authors is not currently state-of-the-art, and there is no comparison with open-set active learning methods. This lack of comparison undermines the validation of the proposed method's effectiveness.

4. The comparison results for OOD detection are predictable. Existing OSAL studies [2,4,6,7] have demonstrated that, due to the limited data in active learning, current OOD detection methods often perform poorly.

[1] Contrastive Coding for Active Learning under Class Distribution Mismatch, ICCV 2021

[2] Active Learning for Open-set Annotation, CVPR 2022

[3] Meta-Query-Net: Resolving Purity-Informativeness Dilemma in Open-set Active Learning, NeurIPS 2022

[4] Not All Out-of-Distribution Data Are Harmful to Open-Set Active Learning, NeurIPS 2023

[5] Entropic Open-set Active Learning, AAAI 2024

[6] Inconsistency-Based Data-Centric Active Open-Set Annotation, AAAI 2024

[7] Bidirectional Uncertainty-Based Active Learning for Open Set Annotation, ECCV 2024

**Questions:**

See weaknesses.

---

### Official Review · Reviewer_8Dpz · 2024-10-28

**Soundness:** 3
**Presentation:** 2
**Contribution:** 2
**Rating:** 3
**Confidence:** 4

**Summary:**

This paper focuses on active learning and out-of-distribution detection. Specifically, the authors aim to propose a unified approach that tries to address OOD detection and active learning simultaneously. The proposal leverages feature space distance metrics to improve the performance. Experimental results on both active learning and OOD detection are reported.

**Strengths:**

1) This paper tries to study a unified approach to address active learning and out-of-distribution detection simultaneously. This problem has rarely been studied.
2) A new distance metrics has been introduced to help improve both the performance of active learning and OOD detection.

**Weaknesses:**

1) Why it is necessary to unify active learning and OOD detection into one framework. How can these tasks help each other? More explanations or theoretical analyses are needed. The contribution of this work to the field of active learning and OOD detection needs to be emphasized.
2) Active learning aims to exploit unlabeled data. Actually, in the field of semi-supervised learning, various works are focusing on open-set or open-world semi-supervised learning, i.e., the unlabeled data contains out-of-distribution instances. The goal is to decrease the negative impact of these OOD unlabeled instances. Can the proposal be applied to this setting?
3) In Table 1, how many runs are the experiments conducted, and how about the performance variance or std?
4) Some other works focus on active learning and OOD[1]. But these works are missing in the related works.

[1] LOG: Active Model Adaptation for Label-Efficient OOD Generalization. NeurIPS 2022.

**Questions:**

As discussed above.

---

### Official Review · Reviewer_3G96 · 2024-11-02

**Soundness:** 1
**Presentation:** 3
**Contribution:** 1
**Rating:** 3
**Confidence:** 5

**Summary:**

This study proposes the Simultaneous Informative Sampling and Outlier Mining (SISOM) methodology, which can be applied to both active learning (AL) and out-of-distribution detection (OoDD) tasks, as they are closely related. Specifically, the method extracts enhanced features by weighting features from all layers with gradients and introduces a scoring scheme based on the distance to classes. The proposed SISOM and its variant combined with energy score, SISOMe, demonstrate comparable or better performance in AL and OoDD settings compared to existing methods.

**Strengths:**

- This study is significant in that it explores the practical applicability of deep learning by simultaneously considering both AL and OoDD tasks. The proposed method is particularly valuable as it is hyperparameter-free, which greatly enhances its practical usability.
- The paper is clearly written and easy to follow. Additionally, the paper includes comparative experiments with many well-known methods in both AL and OoDD tasks.

**Weaknesses:**

- I have concerns about certain overclaims in the study. First, the authors state that they propose the "first unified solution for both AL and OoDD." I understand that this claim is based on the fact that the proposed SISOM score, using a distance ratio, can be applied to both AL and OoDD. However, existing uncertainty metrics, such as MSP and MC Dropout, can also be used as a single metric for both AL and OoDD, making it difficult to assert that this study is the first unified solution. Second, the authors claim that their method achieves top-1 performance on most OOD benchmarks. However, as seen in Table 1, while the method performs well in the Near-OOD setting, it does not do as well in the Far-OOD setting. More precise claims are needed here.
- I am concerned about the computational cost of the proposed method. The SISOM score includes both gradient computation and distance computation between samples, which is not negligible compared to existing methods (even when using a representative subset, $\mathbb{T}$). Additionally, the $r_{avg}$ value in Equation (10) must be updated with each AL cycle, which also requires recalculating the $\alpha_j$ values. These additional computational requirements should be discussed and compared with existing methods.
- I have also concerns regarding the comparison experiments between the proposed method and existing methods. The AL performance shown in Figure 5 does not demonstrate a significant advantage over existing methods. Additionally, I question whether the performance of existing methods in Table 1 has been rigorously evaluated. For example, despite differences in architecture and settings, the performance of ASH on CIFAR-10 differs considerably from the results in the original paper. These points should be clarified to provide readers with sufficient information.

**Questions:**

- In Equation (9), $r_{OOD}$ is proportional to $-r$. Is there a particular reason $r_{OOD}$ needs to be in the form presented in Equation (9)?
- In line 498, it is mentioned that for CIFAR-10, the original set of parameters yielded good performance. How can one determine whether $\alpha$ needs to be optimized? A key advantage of the proposed method is that it is hyperparameter-free, so it would be helpful to provide guidelines on how to decide the value of $\alpha$ from this perspective.

---

### Official Review · Reviewer_RoL4 · 2024-11-04

**Soundness:** 3
**Presentation:** 2
**Contribution:** 2
**Rating:** 3
**Confidence:** 3

**Summary:**

This paper proposes to handle active learning and out-of-distribution detection in a unified framework, SISOM.  SISOM provides
a novel feature space analysis scheme enabling a post-training feature space refinement. By employing enriched feature space distances based on neural coverage technique, SISOM creates a symbiosis between AL and OOD detection.

**Strengths:**

1. This paper proposes SISOM, the first approach designed to jointly solve OOD detection and AL.
2. The experiments on AL and OOD detection are extensive.

**Weaknesses:**

1. In the active learning (AL) experiments, does the unlabeled data include out-of-distribution (OOD) samples? It appears that AL and OOD are discussed as separate scenarios in the experiments. I would like to know if the proposed method can work in a coupled scenario where the unlabeled data contains some OOD samples as noisy data.
2. The improvement in experimental performance is limited.
3. In AL experiments, if 5,000 labeled samples are added in each AL cycle, what is the total number of labeled samples at the end?
4. The data requirements for the AL experiments are not low either. The semi-supervised problem also involves a small number of initial labeled samples and a large amount of unlabeled data, without the need to query for labels. Compared to semi-supervised methods, does the proposed approach offer any performance advantages?

**Questions:**

See weakness.

---

### Note · Authors · 2024-11-14

I have read and agree with the venue's withdrawal policy on behalf of myself and my co-authors.